# Purgative Effect, Acute Toxicity, and Quantification of Phorbol-12-Myristate-13-Acetate and Crotonic Acid in *Croton tiglium* L. Seeds Before and After Treatment by Thai Traditional Detoxification Process

**DOI:** 10.3390/ijms26167714

**Published:** 2025-08-09

**Authors:** Ronnachai Poowanna, Pawitra Pulbutr, Anake Kijjoa, Somsak Nualkaew

**Affiliations:** 1Doctor of Philosophy in Pharmacy Program, Faculty of Pharmacy, Mahasarakham University, Kantharawichai, Maha Sarakham 44150, Thailand; ronnachai.pw@rmuti.ac.th; 2Pharmaceutical Chemistry and Natural Products Research Unit, Faculty of Pharmacy, Mahasarakham University, Kantharawichai, Maha Sarakham 44150, Thailand; pawitra.p@msu.ac.th; 3School of Medicine and Biomedical Sciences (ICBAS) and CIIMAR, Universidade do Porto, Rua de Jorge Viterbo Ferreira, 228, 4050-313 Porto, Portugal; ankijjoa@icbas.up.pt

**Keywords:** *Croton tiglium* seeds, Thai traditional medicine, acute toxicity, purgative activity, phorbol-12-myristate-13-acetate, crotonic acid

## Abstract

*Croton tiglium* L. seeds, a component of many recipes of Thai traditional medicine (TTM), had to undergo the Thai traditional detoxification process (TDP) before being used. However, this detoxification process has never been scientifically proven for its effectiveness. Thus, this research aimed to investigate the effects of TDP on purgative effect and acute toxicity, as well as the identification of some chemical constituents in *C. tiglium* seeds before (CB) and after (CA) treatment. The purgative effect and acute toxicity of CB and CA powders were evaluated using Wistar rats. The amounts of phorbol-12-myristate-13-acetate (PMA) and crotonic acid in the CB and CA powders were determined using HPLC. The results showed no acute toxicity in the rats administered CB and CA powders at doses of 300–2000 mg/kg of body weight (BW). However, CB and CA caused a dose-dependent increase in the number of fecal pellets as well as an increase in the amount of wet and dry feces. Interestingly, only CB, at the dose of 100 mg/kg, caused a significant purgative effect. The TDP was also found to affect the amounts of PMA and crotonic acid. While the amount of PMA in *C. tiglium* seed powder decreased from 1.59 mg/g in CB to 1.26 mg/g in CA, the amount of crotonic acid decreased from 0.001 mg/g in CB to an undetectable level in CA. This investigation demonstrated that TDP not only reduced the purgative effect and toxicity of croton seeds but also the amounts of PMA and crotonic acid.

## 1. Introduction

*Croton tiglium* L. (family Euphorbiaceae), also known as “purging croton” (“salod” in Thai), is a small evergreen tree up to 5–7 m high. This plant is distributed in tropical countries, including Thailand, India, Sri Lanka, China, and Malaysia [1]. The seeds of *C. tiglium* are used in traditional medicine in various countries, both as single and polyherbal recipes, for various purposes such as purgative, flatulence, lymphatic drainage, dyspepsia, and dysentery [2,3,4]. *C. tiglium* seeds have been reported to have a wide range of biological and pharmacological activities, such as antifungal, antibacterial [5], anti-HIV [6], anti-inflammatory [7], and anticancer activities [8,9,10]. There were reports of several anticancer compounds, including alkaloids [9], flavonoids [11,12,13], and diterpenoids, such as 12-*O*-tiglylphorbol-13-(2-methyl) butyrate, 12-*O*-acetylphorbol-13-isobutyrate, 12-*O*-benzoylphorbol-13-(2-methyl) butyrate, 12-*O*-tiglyl-7-oxo-5-ene-phorbol-13-(2-methylbutyrate), and 13-*O*-(2-methyl) butyry1-4-deoxy-4α-phorbol [8] in the ripe seeds of *C. tiglium*.

Although *C. tiglium* seeds had been used in Thai traditional medicine (TTM) for a long time, they have been banned from use as an ingredient in TTM in Thailand since 1978, since they are classified as toxic plant material. Intriguingly, the reason for the prohibition of using *C. tiglium* seeds has never been completely clarified. Chemically, the croton seed oil was reported to contain phorbol esters, crotonic acid, and various fatty acids as major compounds. While the organic acid constituents, such as crotonic acid, were identified as major irritants in the seeds [14], phorbol-12-myristate-13-acetate (PMA) (Figure 1) is an irritant and inflammatory agent that has been extensively used as a tumor promoter on the skin of mice in laboratory settings [15,16]. These findings might be a reason for the prohibition on using *C. tiglium* seeds in traditional medicine in Thailand. Interestingly, Thai traditional medicine practitioners have always been aware of the toxicity and strong purgative effect of *C. tiglium* seeds. For this reason, *C. tiglium* seeds had not been used directly. In order to be able to use them in TTM, the seeds must be subject to the Thai traditional detoxification process (TDP). Not only *C. tiglium* seeds but also several medicinal herbs that possess strong effects or are toxic, such as the latex of *Euphorbia antiquorum* and gamboge, and some elements and inorganic compounds, such as arsenic, borax, realgar (arsenic sulfide), and gamboge, must undergo the TDP to reduce their toxicity or potency in TTM. This method is traditionally known by three different names, *Sa-tu*, *Pra-sa*, and Kha-rith, depending on the type of herb and the terminology passed down through generations. These processes in TTM have been handed down over generations and are roughly described in historical texts [17]. However, the scientific rationale behind these processes is not clearly explained, and their mechanisms have not yet been scientifically proven. The TDP of *C. tiglium* is also known in Thai as “Kharith of Salod”, which means “eliminate the toxicity”. This process was performed by mixing *C. tiglium* seeds, paddy, salt, and water in a clay pot and letting the contents boil over a medium heat until the paddy was disintegrated, after which *C. tiglium* seeds were dried before using them to prepare TTM formulations. TTM practitioners believe that this process can remove the toxicity or decrease the toxic effects of *C. tiglium* seeds by eliminating the toxic compounds to make them safe to use as a valuable drug [14]. Intriguingly, the alteration of chemical constituents, pharmacological activity, and toxicity of *C. tiglium* seeds before and after treatment by TDP has never been previously investigated. Consequently, in this study, we also investigated the effectiveness of TDP in reducing the acute toxicity and purgative activity of *C. tiglium* seeds in the Wistar rat model. The amounts of the main irritant components in *C. tiglium* seeds, i.e., PMA and crotonic acid, were quantified by the HPLC method, and the chromatograms of the seed extracts before and after treatment by TDP were compared to verify the alteration of the constituents. The findings from this study will provide an explanation of using TDP in reducing the toxicity and purgative activity of *C. tiglium* seeds and could lead to a reevaluation of their prohibition from use in traditional medicine in Thailand in the future.

## 2. Results

### 2.1. Acute Toxicity: Sighting Study

The results from the sighting study showed no signs of toxicity, and no rats died in any group from day 1 to day 14. However, the rats in groups 3 and 4, which received 2000 mg/kg of body weight (BW) of CB and CA powders, showed less mobility than the other groups on day 1 and day 2 after receiving the samples. The BW of rats in all groups increased, with groups 3 and 4 showing a slightly lower weight gain compared to the other groups. This discrepancy may be due to differences in food consumption and water intake (Table 1).

### 2.2. Acute Toxicity: Main Study

#### 2.2.1. Physical Examination

The CB powder (group 1) and CA powder (group 2), at the dose of 2000 mg/kg of BW, were tested for acute toxicity in rats and compared with the control group. From the first to the fourteenth day after receiving the samples, no rat in any group died. However, the rats in groups 1 and 2 showed less mobility than the control group in the first two days after receiving the samples.

The weight of rats in all groups increased after 14 d. when compared with the first day. The control group showed the highest increase in BW (20.40 g), followed by the groups that received CA powder (14.26 g) and CB powder (10.79 g), respectively. The BW changes in the CA and CB groups were significantly different from those of the control. Notably, the amount of food consumption and water intake did not show any significant differences among all groups. Specific numerical values and more detailed information are shown in Table 2 and Table 3.

#### 2.2.2. Hematological and Serum Biochemical Parameters

After 14 d of treatment with CB and CA powders, the blood samples were collected from the rat hearts. Hematological parameters and serum biochemical parameters were analyzed.

Most of the rats given CB and CA powders did not show a significant difference from the control group in any of the hematological parameters, including the number of white blood cells (WBC), red blood cells (RBC), hemoglobin concentration (HGB), hematocrit (HCT), mean corpuscular hemoglobin concentration (MCHC), red blood cell distribution width (RDW), mean platelet volume (MPV), mean corpuscular volume (MCV), and mean cell hemoglobin (MCH), except for platelets (PLT), which showed a statistically significant difference. When examining the number of each type of white blood cell, i.e., eosinophils, lymphocytes, monocytes, and neutrophils, the results showed no significant difference when comparing the treatment groups and the control group (Table 4).

There were no significant differences between the treatment groups and the control group in the serum biochemical data, which included blood urea nitrogen [18], uric acid, total protein [19], albumin, total bilirubin (TB), alkaline phosphatase [20], aspartate aminotransferase (AST), alanine transaminase [17], and blood glucose (Table 5).

### 2.3. Purgative Activity

#### 2.3.1. General Data

Before and after the experiment, there was no significant difference in the amount of water and feed consumed by any of the groups. With the exception of group 4 (100 mg/kg CB), which experienced a notable weight loss after 16 h, rats in other groups showed a BW increase. The BW, water, and feed intakes of animals in each group before and after treatment are shown in Figure 2.

#### 2.3.2. Data of Feces

The amounts of wet and dry feces, including water content in the feces in each group, were determined after 8 and 16 h. The amounts of wet feces, dry feces, and feces water content in all groups that received CB and CA powders and the positive control group were higher than those of the negative control group. However, only fecal pellet counts and water content of the positive control group and the group that received 100 mg/kg of CB powder were significantly higher than those of the negative control group. The number and quantity of wet feces, dry feces, and feces water content of animals in each group before and after treatment are shown in Table 6.

#### 2.3.3. Pathological Examination

The pathological examination of the stomach, small intestine, and large intestine tissues was performed after the administration of the sample for 16 h. Figure 3 shows the stomach, small intestine, and large intestine tissues of animals in each group after receiving the samples for 16 h. The submucosa of the stomach tissues of all the animal groups that received CB powder, the groups that received 100 mg/kg CA powder, and the positive control group displayed red blood cells, indicating inflammation of the stomach tissues. Moreover, all groups that received CB powder, CA powder, and the positive control group showed a high number of goblet cells, a cell responsible for mucus secretion, in the small intestine and large intestine when compared to the negative control group. The animal groups that received CB powder tended to have a higher number of goblet cells than the group that received CA powder.

### 2.4. HPLC Chromatogram of CB and CA Extracts

The HPLC chromatograms of CB and CA extracts are shown in Figure 4A,B. The number and intensity of the peaks in the HPLC chromatogram of the CB extract are different from those of the CA extract. Notably, the chromatogram of the CA extract showed a peak, with a retention time of 28 min, which does not exist in the chromatogram of the CB extract.

### 2.5. Determination of the Amounts of Crotonic Acid and PMA

The reference standards, crotonic acid and PMA, showed retention times of 9.45 and 57.55 min, respectively, in the HPLC chromatogram. Then, the peak positions of crotonic acid and PMA in the HPLC chromatograms of the CB and CA extracts were identified by comparison of their retention times with those of the reference standards as well as by spiking crotonic acid and PMA in the extracts. The amounts of crotonic acid and PMA in CB and CA extracts were determined by comparison of the areas under the curve in the HPLC chromatogram with those of the reference standards. The calibration curves of crotonic acid and PMA showed a linear equation of y = 53,628x + 0.4444 and y = 8058.4x − 312.32, respectively. The amount of crotonic acid in the CB powder was 0.001 ± 0.000 mg/g, whereas no crotonic acid was detected in the CA powder. The amount of PMA in the CB powder was 1.59 ± 0.01 mg/g, whereas the amount of PMA in the CA powder was 1.22 ± 0.00 mg/g. The results from the HPLC analyses demonstrated that the TDP decreased the amount of PMA and crotonic acid, the main irritant compounds, in the *C. tiglium* seeds. This HPLC method of analysis was suitable for the simultaneous determination of both crotonic acid and PMA in *C. tiglium* seed extract in a single injection. The validation method of crotonic acid and PMA, the coefficient of determination (R^2^), % relative standard deviation (% RSD) of intraday and interday precision, percent recovery, the limit of detection, and the limit of quantitation passed the standard criteria. The complete data of validation are shown in Table 7.

### 2.6. Identification of Di-(2-ethylhexyl)phthalate (DEHP)

In order to identify the compound that appeared in the HPLC chromatogram of CA (Figure 4B) at a retention time of 28 min, we have made an effort to isolate more quantity of this compound so that its structure can be unraveled by NMR methods. The compound was isolated as a colorless viscous liquid, presenting the (+)-HRESIMS *m*/*z* of 391.2850 (M + H)^+^, calculated for C_24_H_39_O_4_ (391.284972) (Fiure S9). The compound was finally identified as di-(2-ethylhexyl) phthalate (DEHP) (Appendix A) by interpretation of its 1D and 2D NMR spectra (Appendix A) and high-resolution mass spectrum (Appendix A), as well as comparison of its NMR spectral data with those previously reported [21].

## 3. Discussion

*C. tiglium* seeds had been used in TTM for a long time. However, after being classified as a toxic herb by health authorities in Thailand in 1978, *C. tiglium* seeds have been banned from use in TTM until now. Curiously, *C. tiglium* seeds are still used in many other countries [6]. Very interestingly, TTM practitioners have always known about the toxicity and exceedingly strong purgative effect of *C. tiglium* seeds, and therefore, instead of using them directly, the material had to undergo a traditional detoxification process (TDP) (or “Kharith of Salod” in Thai) before being used in TTM formulations [17]. Curiously, this TDP has never been scientifically proven for its validity or efficiency. Thus, the objective of this study was to evaluate the effectiveness of TDP in reducing the purgative effect and toxicity, as well as to determine the extent of the reduction of the amounts of crotonic acid, the major irritant compounds, and PMA, the tumor promoter in *C. tiglium* seeds, after TDP.

The effectiveness of TDP in reducing the purgative effect of *C. tiglium* seeds was verified by comparison of the number of wet feces, dry feces, and water content in feces of Wistar rats that received CB, CA, water (negative control), and castor oil (positive control). The sex, BW, water intake, feed intake, and room atmosphere were maintained in the same condition for each group to control the confounding factors. The results showed that the water intake and the feed intake in all groups were not significantly different before and after the treatment. Almost every group of rats showed BW gain, except for the group that received CB powder at 100 mg/kg, which showed a significant weight loss after 16 h. These results are consistent with the food intake of this group, which consumed significantly less food than the other groups, possibly due to gastrointestinal irritation. The amounts of wet feces, dry feces, and water content in feces after 8 h of the experiment were not significantly different among most groups, except for the group that received CB powder at 100 mg/kg and the positive control group, which had significantly more wet feces than the other groups. After 16 h of treatment, the amounts of wet feces and water content in feces in the group that received CB powder at 100 mg/kg and the positive control group were significantly higher than the other groups. These results are consistent with the previous reports, which indicated that croton oil can stimulate intestinal contractions [22,23]. Croton oil is classified as a stimulant laxative, a type of laxative that usually onsets within 6–12 h, similar to bisacodyl, senna, and sodium picosulfate. Croton oil is known to stimulate nerves that control the muscles in the intestinal wall, helping to induce bowel contractions and facilitate defecation [24]. The rats that received *C. tiglium* seed powder and castor oil showed increased water content in their feces due to the intense stimulation of the intestinal mucosa as well as excessive intestinal contractions by the irritant compounds present in *C. tiglium* seeds. These effects resulted in an increase in intestinal motility, leading to more frequent contractions of the intestines, which reduces the time for nutrient and water absorption from the feces [2]. Consequently, there is a higher amount of residual water in feces than normal. Additionally, the increased secretion of water in the intestines causes the intestinal lining to secrete more water and electrolytes, and the water secreted is not reabsorbed, leading to a higher water content in the feces [25].

Pathological examination of the stomach tissues revealed that the groups receiving all three doses of CB and the group receiving CA powder at 100 mg/kg showed red blood cells in the submucosa, indicating inflammation in the stomach tissues compared to the group receiving only distilled water. Similarly, in the pathological examination of small and large intestine tissues, the groups receiving CB and CA powders showed goblet cells, which are responsible for mucus secretion in the intestines. When the intestines are stimulated, mucus secretion increases. The group receiving CA powder tended to have fewer goblet cells than the group receiving CB powder. The presence of red blood cells in the stomach tissues is often associated with bleeding in the stomach, which may be the result of irritation caused by the compounds present in croton seeds on the gastric mucosa, leading to the formation of gastric ulcers and inflammation. Several studies have indicated that potent chemicals can damage the gastrointestinal lining and stimulate inflammation, which may result in the leakage of blood into the tissues [26,27]. Goblet cells are commonly found in intestinal tissues and play an important role in the production and secretion of mucus, which helps lubricate and protect the intestinal lining from irritation and injury. The increased number of goblet cells in rats that received CB powder is a response of the intestines to the inflammation caused by the toxic effects of CB powder. The relationship between the increase in goblet cells and the inflammatory response is consistent with previous studies on herbal substances that have irritant effects on the gastrointestinal system [28].

The dose of 2000 mg/kg of CB and CA powders was evaluated in the main study for acute toxicity in rats, and their effects were compared with those of the control group. The result showed that no rats died in all groups after receiving the samples from day 1 to day 14. The weight gained by the rats that received CA and CB powders was significantly different when compared to that of the control group, even though the food consumption and water intake were not significantly different. The less weight gained by rats that received CB and CA powders compared to the control group might be due to the purgative and irritation effects of the *C. tiglium* seed powder in the stomach [29].

The size, weight, color, and lesions of the internal organs (brain, heart, thymus, lung, liver, stomach, spleen, left and right adrenal glands, and left and right kidneys) of the rats that received CB and CA powders were not significantly different when compared to the control group. All hematological parameters, including the number of white blood cells, red blood cells, platelets, hemoglobin concentration, hematocrit, mean corpuscular hemoglobin concentration, red blood cell distribution width, and mean platelet volume, mean corpuscular volume, and mean cell hemoglobin, as well as the serum biochemical parameters, including blood urea nitrogen, uric acid, total protein, albumin, total bilirubin, alkaline phosphatase, aspartate aminotransferase, alanine transaminase, and blood glucose of the rats that received CB and CA powders, were not significantly different when compared to the control group. The results of all the experiments in the acute toxicity test indicated that CB and CA powders, at doses up to 2000 mg/kg of BW, were not toxic to rats.

The results of the toxicity study of *C. tiglium* seeds are consistent with those of many previous reports. Toxicological properties of *C. tiglium* seed extract have been previously evaluated by toxicity assays to determine its single-dose acute toxicity (125–2000 mg/kg), 14 d repeated-dose toxicity (125–2000 mg/kg), and 13 wk repeated-dose toxicity (31.25–500 mg/kg) in Sprague–Dawley rats and F344 rats. Hematological, serum biochemical, and histopathological parameters were examined to determine its median lethal dose (LD_50_). The results showed that the acute LD_50_ of *C. tiglium* seed extract in rats was estimated to be greater than 2000 mg/kg. Moreover, the hematological, serum biochemical, and histopathological parameters were not significantly different between the treatment and the control groups [30]. Sub-chronic toxicity of *C. tiglium* seed extract at a dose of 500 mg/kg was also evaluated in both male and female rats, and the results showed that there were no abnormal changes in mortality and behavioral symptoms [30]. EL-Kamali et al. [31], in their study of the toxic effects of crushed *C. tiglium* seeds mixed with animal diet at concentrations of 10% or 20% in male albino rats, have found that oral administration of *C. tiglium* seeds at doses of 10% and 20% not only had a little effect on some hematological indices relating to red blood cells and white blood cells but also showed no significant changes in aspartate aminotransferase and alkaline phosphatase activities between the control and treated animals. On the other hand, the aspartate aminotransferase activity significantly decreased in the group fed a diet containing 20% of *C. tiglium* seeds [31]. The results from these acute and sub-chronic toxicity assessments of *C. tiglium* seeds indicate that both untreated and treated *C. tiglium* seeds appear to be safe via oral administration. It is probable that PMA (and other phorbol esters) can be hydrolyzed in the liver and intestine by enzymes; thus, their tumor-promoter activity could be lost when *C. tiglium* seeds are administered orally [32,33,34]. Given that cancer-promoting activity is a long-term process and depends on many factors, this hypothesis requires further in-depth investigation through long-term studies to confirm its validity.

The carcinogenicity and tumor promotion of *C. tiglium* seeds are the main causes for concern. The mutagenicity of *C. tiglium* seeds was investigated using the Ames test in *Salmonella typhi* TA 98, 100, and 102, and the results showed that it is non-genotoxic in those strains of *S. Typhi* [35]. On the contrary, Kim et al. have found that *C. tiglium* seed extract produced mutagenic responses in five *S. typhimurium* strains in the Ames assay; however, the frequencies of chromosomal aberrations or micronuclei were not altered, indicating that it exerted mutagenic potential and not clastogenicity [36]. Interestingly, there are many reports insisting that *C. tiglium* seed oil is a tumor promoter and not carcinogenic. Most of the previous studies on the tumor-promoting action of croton seed oil were performed on mouse skin. The tumor-promoting activity of croton seed oil on skin carcinogenesis in mice by different chemicals, including benzopyrene, dimethylbenzanthracene, and urethane, has been reported [37]. The tumor-promoting activity of *C. tiglium* seed oil on gastrocarcinogenesis was also investigated. It was found that croton seed oil promoted gastrocarcinogenesis by *N*-methyl-*N*′-nitro-*N*-nitrosoguanidine; however, no tumors were found in rats given only croton seed oil with Tween 60 [38].

The main irritant components of *C. tiglium* seeds are phorbol esters and crotonic acid, as well as some other organic acid constituents. The amount of crotonic acid in *C. tiglium* seed powder before treatment by TDP is 0.001 mg/g, while no crotonic acid was detected after treatment, implying that TDP was able to completely eliminate crotonic acid. The amounts of PMA in *C. tiglium* seed powder decreased from 1.59 mg/g treatment by the TDP to 1.22 mg/g after treatment by the TDP, suggesting that the TDP was, to some extent, able to reduce the amount of PMA. The decrease in the amount of crotonic acid and PMA after the TDP is likely due to the effect of boiling and salt. The decrease in PMA could be due to the hydrolysis of PMA to phorbol. However, at present, there is no available threshold data indicating the specific concentrations of PMA and crotonic acid that would be considered safe or toxic or that would correlate directly with purgative activity in clinical applications. The toxicity or purgative activity of *C. tiglium* seeds may arise from a combination of various phorbol ester compounds. Therefore, while the observed reduction suggests a potential decrease in toxicity, this finding alone cannot conclusively establish the safety or therapeutic effectiveness of the detoxified preparation in clinical contexts. Additional studies, including toxicological and pharmacological evaluations, are required to establish threshold levels and to further assess the safety margins and therapeutic efficacy following TDP.

The isolation of di-(2-ethylhexyl) phthalate (DEHP) from the CA extract is quite intriguing. Di-(2-ethylhexyl) phthalate belongs to the phthalic acid esters (PAEs), a group of synthetic compounds that are widely used as additives in organic solvents and as plasticizers to enhance flexibility and tensile strength in plastics [39]. DEHP can indeed be isolated from various biological sources and is often linked to leaching from plastic containers. DEHP is non-covalently bound to plastics, which can be released into the environment. Interestingly, PAEs, especially DEHP, are also frequently discovered in plant [40,41,42,43] and microorganism [21,44,45] sources, suggesting the possibility that they might be biosynthesized in nature, and they might serve as biologically active substances to enhance competitiveness [46].

Furthermore, the isotope labeling study has demonstrated that PAEs can be biosynthesized by several algae and are presumably stored in cell membranes to keep the flexibility of algal cells [47,48].

The fact that DEHP appeared, in reasonable quantity, in the HPLC chromatogram of the CA extract but not in the CB extract has raised some speculations. The first hypothesis is that DEHP could be derived from the extraction process. However, if this were the case, DEHP should be present in both CA and CB extracts, appearing in both chromatograms since both CA and CB extracts were prepared using the same conditions (solvents and apparatus). Another possible hypothesis is that DEHP could bind to other macromolecules or form complexes with the matrices in the untreated seeds, and are not extractable by the solvent used, but are released from the complex upon boiling with water and salt in the TDP. However, this hypothesis needs to be proved by setting up a suitable model.

When comparing the chromatogram of CB extract (A) with that of the CA extract (B), another interesting feature emerged. The high-intensity peak, with a retention time between 40 and 45 min in the chromatogram of CB extract (A), was sharply diminished in the chromatogram of CA extract (B), indicating a chemical modification or degradation of this compound upon TDP.

This observation highlights an essential aspect of the chemical dynamics in the extracts, which warrants further investigation to understand the implications of these changes on biological activity and safety profiles of the extracts.

It is interesting to mention also that *C. tiglium* seeds are treated before being used in traditional medicine, not only in Thailand but also in other countries. In India, the method of treatment consists of wrapping dried *C. tiglium* seeds in a clean white cloth to form a bolus, which is then soaked in a cow dung solution in a mud pot and boiled. The bolus was then washed with water, and the seeds were treated with cow’s urine, followed by lemon juice. After washing with water, the outer skin and cotyledons were removed, and the seeds were finally fried with ghee. This study was conducted to determine the amount of phorbol and fatty acids, revealing a significant reduction in crotonoside and toxic fatty acids. This method caused a significant reduction of phorbol from 5.18% to 3.86%. Similarly, the amounts of saturated fatty acids, e.g., arachidic acid, behenic acid, stearic acid, and palmitic acid, were also decreased, whereas all the unsaturated fatty acids, such as oleic acid and linoleic acid, seemed to be unaltered [49]. Supporting these findings, Pal, Nandi, and Singh conducted a similar detoxification study using cow’s milk and repeated cycles of steaming and washing. HPLC analysis revealed that phorbol esters decreased from 5.2 to 1.8 mg/100 g, while crotonic acid was completely eliminated after Shodhana [12]. Jamadagni et al. further validated the impact of Shodhana’s method by assessing cytotoxicity and mutagenicity in *C. tiglium* seed extracts. Using MTT assays on Chinese Hamster Ovary (CHO) cells, they have found a marked reduction in IC_50_ values, indicating increased potency after treatment by Shodhana’s method. LC-MS analysis revealed both the disappearance of mutagenic compounds such as phorbol 13-acetate and the emergence of beneficial metabolites like myristic and palmitic acids [35]. Notably, Ames tests showed no significant mutagenic activity up to 2000 mg/plate, affirming the non-genotoxic nature of the purified extracts [35]. On the contrary, no detoxification process of *C. tiglium* seeds was performed before they were used to produce medicine in China [50].

Most studies consistently indicate that *C. tiglium* seeds and PMA are not carcinogenic but act as tumor promoters [36,37,51]. However, cancer caused by PMA requires continuous exposure to carcinogens for a long period of time and at certain concentrations. In addition, numerous studies have reported that not only PMA but also many other phorbol derivatives exhibited anticancer activity [36,52,53,54]. It is probable that these compounds may display dual actions, i.e., promoters of tumor on the one hand, and antagonists of carcinogenesis on the other hand. However, further studies are needed to investigate the chronic toxicity and tumor-promoting effect of *C. tiglium* seeds after being treated with TDP.

## 4. Materials and Methods

### 4.1. Plant Material

*C. tiglium* seeds (Figure 5D) were collected in July 2021 from Chumphon Province, Thailand, located between the latitudes of 10°29′ N and longitudes of 98°11′ E. The plant material was identified by Prof. Wanida Caichompoo, from the Faculty of Pharmacy, Mahasarakham University. The voucher specimen (voucher No. MSU.PH-EUP-C1) was deposited at the herbarium of the Faculty of Pharmacy, Mahasarakham University, Maha Sarakham, Thailand.

### 4.2. Chemicals and Reagents

The reference standards, phorbol-12-myristate-13-acetate (PMA) and crotonic acid, were purchased from Merck (Darmstadt, Germany). All other chemicals used were of analytical grade. The HPLC-grade solvents were used for HPLC analysis. Methanol, trifluoroacetic acid, and isoflurane were purchased from Merck (Darmstadt, Germany). Salt (a food grade) was purchased from Kudreakum Sakon Nakhon, Thailand.

### 4.3. Treatment of C. tiglium Seeds by Thai Traditional Detoxification Process (TDP)

*C. tiglium* seeds (50 g), paddy (150 g), salt (100 g), and water (2 L) were added to the clay pot and boiled over a medium heat (150 °C) until the paddy was disintegrated (ca. 4 h), and then filtered. The solid residue was rinsed with water thoroughly and subsequently dried in the hot-air oven at 40 °C for 48 h. This process is called “Kharith of Salod” in the Thai language.

### 4.4. Preparation of Croton Seeds

#### 4.4.1. For Acute Toxicity Test and Purgative Activity

*C. tiglium* seeds before treatment (CB) and after treatment (CA) by TDP were ground, using a laboratory mill (Spring Green Evolution, Bangkok, Thailand), to a fine powder and kept in a well-closed container at −20 °C until use.

#### 4.4.2. For Determination of Phorbol-12-Myristate-13-Acetate (PMA)

*C. tiglium* seeds (CB and CA) were ground to a fine powder using the same method described above. Each *C. tiglium* seed powder (25 g) was macerated in methanol (150 mL) and sonicated for 30 min. The content was filtered, and the solution was evaporated to dryness by a rotary evaporator to obtain crude extracts of CB (50 mg) and CA (24 mg). The extracts were kept in a well-closed container at −20 °C until use.

### 4.5. Acute Toxicity

#### 4.5.1. Animals

Adult female Wistar rats weighing ca. 150–200 g and about 6 weeks old, obtained from the national laboratory animal center (Mahidol University, Bangkok, Thailand), were acclimatized at the Northeast Laboratory Animal Center, Khon Kaen University, Khon Kaen, Thailand, in an air-conditioned room at 25 ± 2 °C, 12-h light/12-h dark cycle, with relative humidity of 40–60%. Animals were housed in metabolic cages (3 Wistar rats per cage) with standard chow and water *ad libitum* for 7 d prior to commencing experiments. The rats were fasted for 18 h before starting the experiments. All experiments were carried out according to the Committee of Care and Use of Laboratory Animal Resources, the National Research Council of Thailand, and the Organization for Economic Cooperation and Development guidelines for testing of chemicals number 420 [55] and performed following the advice of The Institutional Animal Care and Use Committee of Khon Kaen University, Thailand (approval number 660201.2.11/122 (30)).

#### 4.5.2. Acute Toxicity Test

The fixed-dose study was conducted following the recommendations of the OECD Guidelines for the Testing of Chemicals. This method was divided into two phases—the sighting study and the main study. The acute toxicity studies were carried out in seven-week-old healthy female Wistar rats with 150–200 g of BW, using a single-dose oral administration.

#### 4.5.3. Sighting Study

A sighting study is an initial test performed to determine the initial dose of the substance that will be used in the main study for toxicity testing. Rats were divided into five groups, with one rat per group. After fasting overnight, groups 1 and 2 received 300 mg/kg of BW of CB and CA powders, respectively. Groups 3 and 4 received 2000 mg/kg of BW of CB and CA powder, respectively, while the control group received distilled water. Animals were observed for toxic manifestations for 24 h after treatment. The number of dead animals was recorded. The surviving animals were observed for another 14 d. At the end of the experiment, all rats were euthanized by isoflurane (2–5%) inhalation, and an autopsy was performed to detect any abnormalities.

#### 4.5.4. Main Study

Since the doses of 300 and 2000 mg/kg of BW did not show any sign of toxicity in the sighting study, the dose of 2000 mg/kg of BW was chosen for investigation in the main study. Rats were divided into three groups of five rats per group (*n* = 5). Groups 1 and 2 were fed with 2000 mg/kg of BW of CB and CA powders, respectively, while the control group was treated with distilled water. Animals were observed for toxic manifestations for 24 h after treatment. The number of dead animals was recorded. The surviving animals were observed for another 14 d. At the end of the experiment, all rats were euthanized by isoflurane (2–5%) inhalation. Blood samples were collected from all animal groups and examined using a blood analyzer (Sysmex, XN-1000, Sysmex Corporation, Kobe, Japan, for a complete blood count test and Roche, Cobas Integra 400 plus, Roche Diagnostics, Basel, Switzerland, for a blood chemistry test). The hematological and biochemical parameters of the serum were analyzed immediately after separation from whole blood samples. Reference values of hematological and biochemical parameters of animals were the same as those in the previous reports [20,56,57].

### 4.6. Purgative Activity

#### 4.6.1. Animals

Six-week-old male Wistar rats weighing between 150 and 200 g were acquired from Khon Kaen University’s Northeast Laboratory Animal Center. The animals were housed individually in clean metabolic cages with filter paper placed under the cages to collect each rat’s feces in a well-ventilated house under optimum conditions (23 ± 1 °C, 12 h natural light and 12 h dark, with relative humidity of 45–50%). The rats were acclimatized for 7 d and fasted for 18 h before starting the experiments, during which they were allowed free access to commercial pelleted rat chow and distilled water. All animal treatments were in accordance with international ethical guidelines and the National Institute of Health Guide concerning the care and use of laboratory animals. The study was carried out following the approval from the Ethical Committee of the Institution Animal Care and Use Committee of Khon Khen University, Record No. IACUC-KKU-28/64, Reference No. 660201.2.11/122 (30).

#### 4.6.2. Purgative Activity Test

The purgative activity test was conducted according to the method previously described by Meite et al. [58], with some modifications. Briefly, a total of 48 Wistar rats, divided into 8 groups of 6 animals per group, were used in the experiment. Rats in groups 1 and 2 received distilled water (negative control) and castor oil (PC, 0.3 mL/animal), respectively. CB and CA powders at doses of 10, 50, and 100 mg/kg of BW were given to rats in groups 3–8, respectively. The dose of CB and CA powders was calculated based on the average amount found in TTM recipes from Thai traditional medicine textbooks, where the typical range is 10–60 mg/kg of BW. The higher dose level of 100 mg/kg of BW was also included. Water intake, feed intake, and BW gain were recorded throughout the experimental period, with treatments lasting for 8 and 16 h. Fecal pellets from each rat were collected daily in the morning and throughout the experiment. The total number, weight, and water content of the pellets were recorded. Water content was calculated by determining the difference between the wet and dry weights of the pellets. At the end of the experiment, all rats were euthanized by isoflurane (2–5%) inhalation, and the stomach and intestinal tissues of the test animals were dissected and stained with hematoxylin and eosin to observe the pathological changes in the respective tissues.

### 4.7. Determination of Toxic Compounds in the Seed Extracts

#### 4.7.1. Determination of Crotonic Acid and PMA

The amounts of crotonic acid and PMA in CB and CA extracts were quantified by the HPLC method. CB and CA extracts were separately dissolved in MeOH to a final concentration of 2 mg/mL and then subjected to HPLC for both qualitative and quantitative analyses. The HPLC system consists of an Agilent 1260 Infinity II Analytical-Scale LC Purification System, which was equipped with a diode array detector (Agilent 1260 Infinity II, Santa Clara, CA, USA), an LC column (Phenomenex, Luna 5 µm (RP-C18) 250 X 4.6 nm, a Guard column (Phenomenex 4.0), and the data were processed by Agilent HPLC-LC1260 openlab CDS chemstation edition software. Separation was achieved with a linear gradient program using H_2_O containing 0.05% trifluoroacetic acid (solvent A) and MeOH (solvent B) as a mobile phase. The elution was performed using a gradient system of A:B from 90:10 (3 min), 75:25 (7 min), 60:40 (15 min), 30:70 (10 min), 20:80 (25 min), 10:90 (35 min), and 100% (17 min). The flow rate was 0.8 mL/min, and the detector was set at 254 nm. The amounts of crotonic acid and PMA in CB and CA extracts were determined by comparison of the peak areas in the chromatograms of the samples with those of the calibration curve of the reference standards. Each experiment was performed in three replicates at room temperature. The HPLC fingerprints of CB and CA extracts were also compared.

#### 4.7.2. Validation Method

The method for quantification of crotonic acid and PMA by HPLC, including linearity, the limit of detection, the limit of quantitation, precision, and accuracy, was validated prior to the determination of crotonic acid and PMA in CB and CA extracts, as follows.

##### Linearity

Crotonic acid and PMA were prepared at various concentrations and analyzed by HPLC under the same conditions described above. A calibration curve was constructed by plotting PMA concentrations (x-axis) against the area under the curve (y-axis), and linear regression analysis was performed.

##### Limit of Detection (LOD) and Limit of Quantitation (LOQ)

Small amounts of low concentration of crotonic acid and PMA were added to the sample and analyzed by HPLC in ten replicates. The standard deviations of crotonic acid and PMA were calculated. The LOD was defined as 3 times the standard deviation (3 SD), and the LOQ as 10 times the standard deviation (10 SD).

##### Precision

Three concentrations (high, medium, and low) of crotonic acid and PMA were analyzed by HPLC three times on the same day (intraday) and over three consecutive days (interday). Each experiment was performed in triplicate, and the percent relative standard deviation (% RSD) was calculated.

##### Accuracy

Three concentrations (high, medium, and low) of crotonic acid and PMA were added to the sample and analyzed by HPLC. Each experiment was performed in seven replicates, and the percent recovery was calculated.

### 4.8. Isolation of Di-(2-ethylhexyl)phthalate (DEHP)

Di-(2-ethylhexyl) phthalate (DEHP) was first observed in the HPLC chromatogram of CA (B) at a retention time of 28 min. The CA extract (10 g) was applied on a silica gel (100–200 mesh) column and eluted successively with a gradient of CHCl_2_-EtOAc (from 8:2 to 0:10), followed by EtOAc:MeOH (from 9:1 to 6:4) to give 13 fractions (Frs). Frs 1 (1.6 g) and 2 (1.08 g) were combined and applied on a column chromatography using resin-20 as a stationary phase and eluted successively with a gradient of H_2_O:MeOH (from 10:0 to 0:10) to obtain 12 frs. Frs. 10 (11 mg) and 11 (8 mg) were combined and dissolved in CHCl_2_, and the solution was evaporated to dryness to give pure DEHP (17 mg).

### 4.9. Structure Elucidation of DEHP

Pure DEHP was dissolved in CDCl_3_ for NMR analysis. ^1^H and ^13^C NMR spectra were recorded at ambient temperature on a Bruker Ascend spectrometer (Bruker Biospin AG, Fällanden, Switzerland) operating at 400 and 100 MHz, respectively. High-resolution mass spectra were measured with a Bruker Compact QToF mass spectrometer (Bruker Daltonik, Bremen, Germany).

### 4.10. Statistical Analysis

Individual data were examined by one-way analysis of variance (ANOVA) using SPSS software version 19 (IBM, Chicago, IL, USA), and multiple comparisons were made using Dunnett’s test (SPSS Inc., Chicago, IL, USA). The threshold for statistical significance was fixed at *p* < 0.05.

## 5. Conclusions

The present study showed that *C. tiglium* seeds have no acute toxicity in rats when administered orally. The Thai traditional detoxification process (TDP) used by Thai traditional medicine practitioners can not only completely eliminate crotonic acid and partially decrease the amount of PMA, well-known irritant constituents of *C. tiglium* seeds, but also decrease other unidentified compounds. This investigation also demonstrated that TDP reduced the purgative effect and toxicity of *C. tiglium* seeds. These findings may warrant the TDP of *C. tiglium* seeds as a method to guarantee the safety of *C. tiglium* seeds for use in Thai traditional medicine, especially in polyherbal recipes. The results obtained from this study are relevant to promote the discussion to reevaluate the status of *C. tiglium* seeds as a toxic herb and their prohibition from use in Thai traditional medicine.

## Figures and Tables

**Figure 1 ijms-26-07714-f001:**
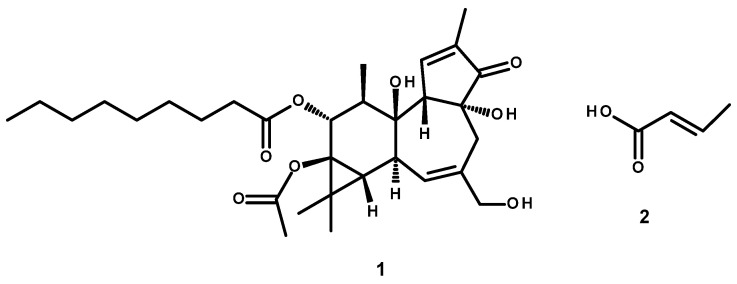
Chemical structure of phorbol-12-myristate-13-acetate (**1**) and crotonic acid (**2**).

**Figure 2 ijms-26-07714-f002:**
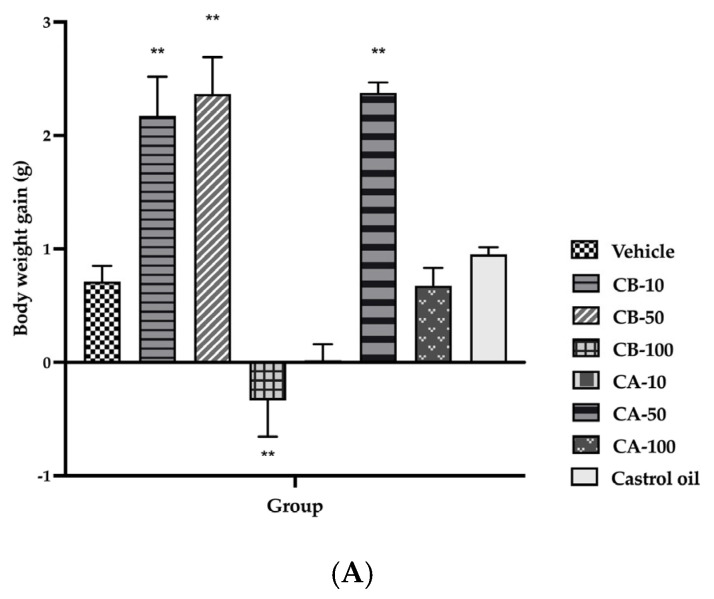
The purgative effect of CB and CA powders on BW gain (**A**), water intake (**B**), and feed intake (**C**) of Wistar rats (*n* = 6) at the doses of 10, 50, and 100 mg/kg compared to the negative control (vehicle) and the positive control (castor oil) after 16 h observation, ** *p* < 0.01.

**Figure 3 ijms-26-07714-f003:**
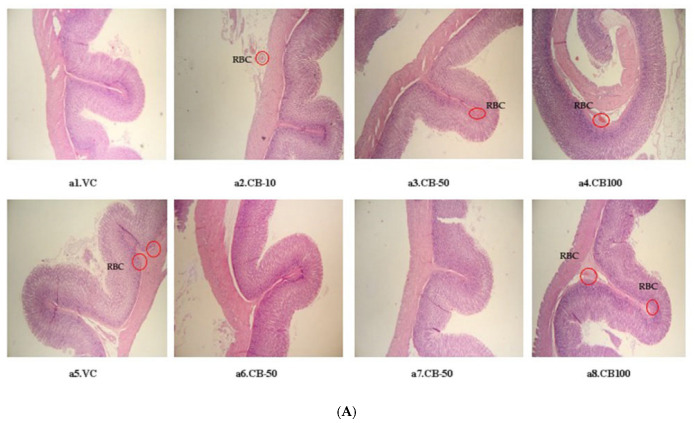
The purgative effect of CB and CA powder on cross-section of the stomach (**A**), large intestine (**B**), and small intestine (**C**) tissues in Wistar rats after receiving: (**a1**,**b1**,**c1**) vehicle, (**a2**,**b2**,**c2**) castor oil, (**a3**,**b3**,**c3**) CB powder 10 mg/kg (CB-10), (**a4**,**b4**,**c4**) CB powder 50 mg/kg (CB-50), (**a5**,**b5**,**c5**) CB powder 100 mg/kg (CB-100), (**a6**,**b6**,**c6**) CA powder 10 mg/kg (CA-10), (**a7**,**b7**,**c7**) CA powder 50 mg/kg (CA-50), and (**a8**,**b8**,**c8**) CA powder 100 mg/kg (CA-100), RBC = red blood cells, GB = goblet cell.

**Figure 4 ijms-26-07714-f004:**
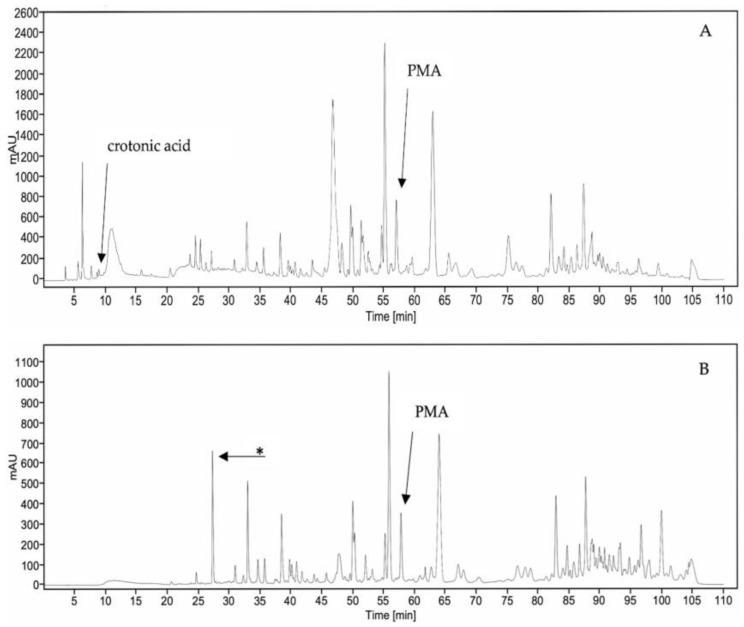
HPLC chromatogram of *C. tiglium* seed extract before treatment (CB) (**A**) and after treatment (CA) (**B**) by TDP (* Chemical artifact).

**Figure 5 ijms-26-07714-f005:**
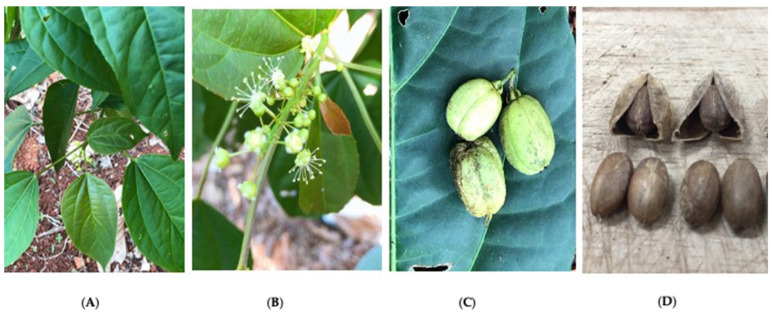
Photographs of *C. tiglium* L., leaves (**A**), flowers (**B**), fresh fruits (**C**), and dry fruits and seeds (**D**).

**Table 1 ijms-26-07714-t001:** The acute toxicity: a sighting study of CB and CA powders on body weight (BW), food consumption, and water intake of Wistar rats at the doses of 300 and 2000 mg/kg compared to a negative control (vehicle).

Group	Dose (mg/kg)	Body Weight (BW) (g)			Food Consumption (g)		Water Intake (g)	
		Day 0	Week 1	Week 2	Week 1	Week 2	Week 1	Week 2
1 (CB)	300	203.12 ± 2.54	221.12 ± 3.14	234.53 ± 3.40	65.12 ± 2.81	115.42 ± 2.91	200.25 ± 4.35	448.35 ± 4.35
2 (CA)	300	194.38 ± 2.41	214.38 ± 2.44	230.39 ± 3.21	68.24 ± 1.51	121.52 ± 1.35	214.55 ± 7.05	454.25 ± 3.35
3 (CB)	2000	208.54 ± 3.54	225.10 ± 3.54	235.22 ± 2.41	65.35 ± 1.21	113.12 ± 3.11	156.25 ± 7.05	360.65 ± 2.15
4 (CA)	2000	195.02 ± 1.54	215.28 ± 3.41	222.22 ± 4.41	68.14 ± 1.11	85.22 ± 2.35	144.55 ± 6.25	340.85 ± 6.75
5 (Control)		202.11 ± 3.54	220.02 ± 2.51	234.80 ± 2.81	60.04 ± 1.91	125.82 ± 4.05	220.15 ± 4.15	480.25 ± 4.50

**Table 2 ijms-26-07714-t002:** The acute toxicity: a main study of CB and CA powders on the body weight of Wistar rats at the dose of 2000 mg/kg compared to the negative control (vehicle).

Group	BW (g)			BW Change (g)
	Day 0	Week 1	Week 2	
1 (CB)	230.31 ± 5.05 ^a^	231.04 ± 3.60 ^a^	241.10 ± 5.02 ^a^	10.79 ± 0.54 ^b^
2 (CA)	232.71 ± 5.17 ^a^	239.80 ± 4.35 ^a^	246.97 ± 4.85 ^a^	14.26 ± 2.12 ^b^
3 (Control)	224.49 ± 4.78 ^a^	233.77 ± 6.25 ^a^	244.89 ± 5.42 ^a^	20.40 ± 2.39 ^a^

The values represent the mean ± SEM (*n* = 5). Within the same column, different superscript letters indicate significant differences at *p* < 0.05.

**Table 3 ijms-26-07714-t003:** The acute toxicity: a main study of CB and CA powders on food consumption and water intake of Wistar rats at the dose of 2000 mg/kg compared to the negative control (vehicle).

Group	Food Consumption (g)		Water Intake (g)	
	Week 1	Week 2	Week 1	Week 2
1 (CB)	14.00 ± 0.53 ^a^	17.02 ± 0.46 ^a^	33.31 ± 5.76 ^a^	33.11 ± 3.26 ^a^
2 (CA)	17.00 ± 2.42 ^a^	17.00 ± 1.27 ^a^	34.10 ± 8.14 ^a^	30.31 ± 2.30 ^a^
3 (Control)	17.71 ± 0.61 ^a^	17.88 ± 0.41 ^a^	29.51 ± 3.21 ^a^	28.86 ± 2.21 ^a^

The values represent the mean ± SEM (*n* = 5). Within the same column, different superscript letters indicate significant differences at *p* < 0.05.

**Table 4 ijms-26-07714-t004:** The acute toxicity: a main study of CB and CA powders on hematological parameters of Wistar rats at the dose of 2000 mg/kg after 14 d compared to a negative control (vehicle).

Hematological Parameters	Group		
	1 (CB)	2 (CA)	Control
WBC (10^3^/μL)	2.66 ± 0.34 ^a^	2.52 ± 0.21 ^a^	3.16 ± 0.32 ^a^
RBC (10^6^/μL)	7.43 ± 0.11 ^a^	7.15 ± 0.15 ^a^	7.27 ± 0.11 ^a^
HGB (g/dL)	13.40 ± 0.13 ^a^	13.00 ± 0.33 ^a^	13.46 ± 0.27 ^a^
HCT (%)	38.78 ± 0.48 ^a^	37.97 ± 0.67 ^a^	39.16 ± 0.75 ^a^
PLT (10^3^/μL)	976.20 ± 49.92 ^a^	1018.10 ± 140.88 ^b^	907.40 ± 41.49 ^a^
MCV (fL)	52.20 ± 0.54 ^b^	53.05 ± 0.75 ^a^	53.84 ± 0.27 ^a^
MCH (pg)	18.04 ± 0.11 ^b^	18.15 ± 0.18 ^a^	18.50 ± 0.11 ^a^
MCHC (g/dL)	34.56 ± 0.22 ^a^	34.22 ± 0.37 ^a^	34.36 ± 0.09 ^a^
RDW-SD (%)	25.90 ± 0.44 ^a^	26.35 ± 0.43 ^a^	26.68 ± 0.66 ^a^
MPV (fL)	8.28 ± 0.18 ^a^	8.55 ± 0.30 ^a^	7.96 ± 0.12 ^a^
Neutrophils (%)	9.16 ± 2.06 ^a^	9.47 ± 0.87 ^a^	11.20 ± 1.14 ^a^
Eosinophils (%)	1.84 ± 0.46 ^a^	1.70 ± 0.57 ^a^	2.22 ± 0.48 ^a^
Lymphocytes (%)	86.84 ± 3.11 ^a^	85.30 ± 4.81 ^a^	84.46 ± 1.70 ^a^
Monocytes (%)	2.16 ± 0.65 ^a^	2.32 ± 0.57 ^a^	2.12 ± 0.40 ^a^

The values represent the mean ± SEM (*n* = 5). Within the same row, different superscript letters indicate significant differences at *p* < 0.05.

**Table 5 ijms-26-07714-t005:** The acute toxicity: a main study of CB and CA powders on biochemical parameters of Wistar rats at the dose of 2000 mg/kg after 14 d compared to a negative control (vehicle).

Biochemical Parameters	Group		
	1 (CB)	2 (CA)	Control
BUN (mg/dL)	21.20 ± 1.24 ^a^	25.00 ± 2.85 ^a^	21.80 ± 0.91 ^a^
Uric Acid (mg/dL)	1.42 ± 0.13 ^a^	1.37 ± 0.17 ^a^	1.20 ± 0.08 ^a^
TP (g/dL)	6.04 ± 0.16 ^a^	5.62 ± 0.07 ^a^	5.80 ± 0.19 ^a^
Albumin (g/dL)	4.62 ± 0.19 ^a^	4.07 ± 0.13 ^a^	4.40 ± 0.13 ^a^
TB (mg/dL)	0.10 ± 0.00 ^a^	0.10 ± 0.00 ^a^	0.10 ± 0.00 ^a^
ALP (IU/L)	72.20 ± 5.13 ^a^	81.50 ± 12.55 ^a^	79.60 ± 8.29 ^a^
AST (IU/L)	77.20 ± 2.35 ^a^	91.00 ± 7.90 ^a^	76.20 ± 5.34 ^a^
ALT (IU/L)	27.60 ± 2.97 ^a^	22.00 ± 1.22 ^a^	26.60 ± 1.69 ^a^
Glucose (mg/dL)	174.40 ± 6.40 ^a^	176.75 ± 6.35 ^a^	176.60 ± 4.54 ^a^

The values represent the mean ± SEM (*n* = 5). Within the same row, different superscript letters indicate significant differences at *p* < 0.05.

**Table 6 ijms-26-07714-t006:** The purgative effect of CB and CA powders on the number of feces in Wistar rats at the doses of 10, 50, and 100 mg/kg compared to the negative control (vehicle) and the positive control (castor oil).

Group	Number of Feces (Mean ± SEM)	Amount of Wet Feces (Mean ± SE) (g)	Amount of Dry Feces (Mean ± SE) (g)	% of Feces Water Content (Mean ± SE)
Vehicle	16.67 ± 0.80	3.57 ± 0.17	2.15 ± 0.06	34.81 ± 2.97
CB-10	16.50 ± 1.73	4.05 ± 0.45	2.26 ± 0.26	44.28 ± 2.47
CB-50	17.50 ± 2.57	4.57 ± 0.48	2.34 ± 0.19	44.30 ± 2.23
CB-100	18.17 ± 1.33 *	5.31 ± 0.29 *	2.40 ± 0.15	54.63 ± 2.39 **
CA-10	15.67 ± 1.28	3.62 ± 0.33	2.14 ± 0.29	41.72 ± 3.72
CA-50	16.50 ± 0.67	3.81 ± 0.36	2.24 ± 0.31	42.48 ± 4.58
CA-100	15.67 ± 1.02	4.07 ± 0.29	2.25 ± 0.24	46.00 ± 4.32
Castor oil	19.83 ± 0.87 *	4.66 ± 0.25 *	2.33 ± 0.10	49.61 ± 2.75 *

The values represent the mean ± SEM, significant differences at * *p* < 0.05 and ** *p* < 0.01 (*n* = 5).

**Table 7 ijms-26-07714-t007:** Validation method for the determination of crotonic acid and phorbol-12-myristate-13-acetate (PMA) in CB and CA by HPLC.

Parameters	Crotonic Acid	PMA
Linearity range (µg/mL)	1–100	1–500
Precision (% RSD)		
- Intraday precision	0.98–1.95	1.34–5.68
- Interday precision	1.03–1.91	3.89–4.82
Accuracy (% recovery)	99.79–104.08	101.09–102.08
LOD (µg/mL)	0.98	0.94
LOQ (µg/mL)	3.27	3.12

## Data Availability

The original contributions presented in this study are included in the article. Further inquiries can be directed to the corresponding author.

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
