# Peer review of "Purgative Effect, Acute Toxicity, and Quantification of Phorbol-12-Myristate-13-Acetate and Crotonic Acid in Croton tiglium L. Seeds Before and After Treatment by Thai Traditional Detoxification Process"

_ijms, 2025, doi:10.3390/ijms26167714_

Round 1
Reviewer 1 Report (Previous Reviewer 2)
Comments and Suggestions for Authors
I reviewed the manuscript entitled Purgative Effect, Acute Toxicity, and Quantification of Phorbol-12-Myristate-13-Acetate and Crotonic Acid in Croton tiglium L. Seeds before and after Treatment by Thai Traditional Detoxification Process.
I agree to accept this manuscript after major revision.
1) The paper mentions that the scientific effectiveness of Thailand's traditional detoxification process (TDP) has never been verified, but does not provide a detailed explanation for why this process was chosen as the research object. Can you provide more information on the historical application background and importance of TDP in traditional Thai medicine?
2) Why are the doses of 300 mg/kg and 2000 mg/kg chosen in acute toxicity experiments? Is there any previous research supporting the selection of these doses? Have you considered a wider range of doses to comprehensively evaluate toxicity?
3) The paper points out that TDP reduces the levels of PMA and crotonic acid, but does not discuss whether this reduction is sufficient to ensure the safety or effectiveness of clinical applications. Is there threshold data to support the toxicological significance of this reduction?
4) DEHP was found in CA samples, but not detected in CB. The paper suggests that DEHP may come from the TDP process, but does not rule out the possibility of experimental contamination. Does the author need further experimental verification of its source (such as blank control)?
5) The paper only evaluated acute toxicity, but the potential risks of PMA as a tumor promoter may require long-term toxicity experiments. Have you considered supplementing subchronic or chronic toxicity studies?
6) In the Purgative Activity Test, castor oil was used as the positive control, but the basis for its dosage selection was not specified. Is there any literature supporting comparability between this dose and the purgative effect of croton seeds?
7) The paper mentions that the HPLC method has been validated, but does not provide specific data on key parameters such as linear range and recovery rate. Can these data be fully listed in the supplementary materials?
8) The description of inflammation in gastric and intestinal tissues, such as red blood cell infiltration and goblet cell count, lacks quantitative analysis. Can more objective data be provided through image analysis software?
9) The paper mentions that PMA may lose its activity through gastric acid hydrolysis, but does not cite relevant literature to support this hypothesis. Can the author provide additional experimental or literature evidence to simulate the gastric acid environment in vitro?
Author Response
Comments 1:The paper mentions that the scientific effectiveness of Thailand's traditional detoxification process (TDP) has never been verified, but does not provide a detailed explanation for why this process was chosen as the research object. Can you provide more information on the historical application background and importance of TDP in traditional Thai medicine? |
Response 1: We would like to thank reviewer#1 for his/her constructive comment. More information of TDP was added in the revised manuscript (Page 2, lines 63-71). The information added was highlighted in red as follow “Not only C. tiglium seeds, but several medicinal herbs that possess strong effects or are toxic such as latex of Euphorbia antiquorum, and gamboge, and some elements and inorganic compounds such as arsenic, borax, realgar (arsenic sulfide), must undergo the TDP to reduce their toxicity or potency in TTM. This method is traditionally known by three different names, Sa-tu, Pra-sa, and Kha-rith, depending on the type of herb and the terminology passed down through generations. These processes have been handed down over generations and are roughly described in historical texts [17]. However, the scientific rationale behind these processes is not clearly explained, and their mechanisms have not yet been scientifically proven.” |
Comments 2: Why are the doses of 300 mg/kg and 2000 mg/kg chosen in acute toxicity experiments? Is there any previous research supporting the selection of these doses? Have you considered a wider range of doses to comprehensively evaluate toxicity? |
Response 2: We calculated the dose of C. tiglium seeds based on the average amount found in Thai traditional medicine recipes from Thai traditional medicine textbooks, where the typical range is 10–60 mg/kg BW (Page 17, lines 559-561). These doses are relatively safe; therefore, we are interested in investigating doses higher than the normal level, ranging from 5 times the typical dose up to the maximum dose that can be administered to rats. |
Comments 3: The paper points out that TDP reduces the levels of PMA and crotonic acid, but does not discuss whether this reduction is sufficient to ensure the safety or effectiveness of clinical applications. Is there threshold data to support the toxicological significance of this reduction? |
Response 3: We wish to thank reviewer#1 for his/her insightful comment. As mentioned, our study demonstrated that the TDP significantly reduces the levels of PMA and crotonic acid in C. tiglium seeds. However, at present, there is no available threshold data indicating the specific concentrations of PMA and crotonic acid that would be considered safe or toxic, or that would correlate directly with purgative activity in clinical applications. The toxicity or purgative activity of C. tiglium seeds may arise from a combination of various phorbol esteres. Therefore, while the observed reduction suggests a potential decrease in toxicity, this finding alone cannot conclusively establish the safety or therapeutic effectiveness of the detoxified preparation in clinical contexts. Additional studies, including toxicological and pharmacological evaluations, are required to establish threshold levels and to further assess the safety margins and therapeutic efficacy following TDP. We have now added this clarification to the discussion section of the manuscript to highlight the need for further investigation (Page 13, lines 382-391). |
Comments 4: DEHP was found in CA samples, but not detected in CB. The paper suggests that DEHP may come from the TDP process, but does not rule out the possibility of experimental contamination. Does the author need further experimental verification of its source (such as blank control)? |
Response 4: We have already addressed this issue in the discussion section. “The first hypothesis is that DEHP could derive from the extraction process. However, if this was the case, DEHP should be present in both CA and CB extracts, appearing in both chromatograms since both CA and CB extracts were prepared using the same condition (solvents and apparatus). Another possible hypothesis is that DEHP could bind to other macromolecules or form complexes with the matrices in the untreated seeds and are not extractable by the solvent used, but is released from the complex upon boiling with water and salt in the TDP. However, this hypothesis needs to be proved by setting up a suitable model” (Page 13, lines 405-413). |
Comments 5: The paper only evaluated acute toxicity, but the potential risks of PMA as a tumor promoter may require long-term toxicity experiments. Have you considered supplementing subchronic or chronic toxicity studies? |
Response 5: We thank reviewer#1 for this interesting comment. Certainly, while our study focused on acute toxicity, PMA is known to possess tumor-promoting properties, which may pose risks associated with long-term exposure. Therefore, evaluation of subchronic or chronic toxicity is indeed essential for a comprehensive safety assessment. At this stage, our aim was to conduct a preliminary evaluation of chemical composition changes and acute toxicity following the TDP as a foundational study. We fully agree that further studies are necessary, and we have planned to conduct subchronic and chronic toxicity experiments in future research to address the long-term safety concerns of detoxified C. tiglium products. This limitation and future direction have been mentioned in the discussion section of the manuscript (Page 14, lines 452-454). |
Comments 6: In the Purgative Activity Test, castor oil was used as the positive control, but the basis for its dosage selection was not specified. Is there any literature supporting comparability between this dose and the purgative effect of croton seeds? |
Response 6: In studies evaluating the purgative (laxative) activity of castor oil in rats, the typical dose range is:1 to 2 mL/kg, administered orally. This dose is commonly used to induce diarrhea or assess the effect of other substances (e.g., antidiarrheal or laxative agents) in experimental models (please find the references below). Since the average body weight of the rats used in this experiment was 150 g (0.15 kg), thus the castor oil dose given was 0.15 kg * 2 mL = 0.3 mL/ rat.
The following is a list of references for castor oil dosage: 1. Méité S, Bahi C, Yéo D, Datté JY, Djaman JA, N'guessan DJ. Laxative activities of Mareya micrantha (Benth.) Müll. Arg. (Euphorbiaceae) leaf aqueous extract in rats. BMC Complement Altern Med. 2010 Feb 16;10:7. doi: 10.1186/1472-6882-10-7. PMID: 20158903; PMCID: PMC2830176. 2. Meite et al., Tropical Journal of Pharmaceutical Research, June 2009; 8 (3): 201-207. 3. Capasso F, Mascolo N, Izzo AA, Gaginella TS. Dissociation of castor oil-induced diarrhoea and intestinal mucosal injury in rat: effect of NG-nitro-L-arginine methyl ester. Br J Pharmacol. 1994 Dec;113(4):1127-30. doi: 10.1111/j.1476-5381.1994.tb17113.x. PMID: 7889264; PMCID: PMC1510485. |
Comments 7: The paper mentions that the HPLC method has been validated, but does not provide specific data on key parameters such as linear range and recovery rate. Can these data be fully listed in the supplementary materials? |
Response 7: We appreciate reviewer#1’s suggestion regarding the HPLC method validation. We initially overlooked the linear range; however, the % recovery has already been provided in Table 1. We have now added the linear ranges of PMA and crotonic acid concentration in Table 1 of the revised manuscript. (Page 10, line 238). |
Comments 8: The description of inflammation in gastric and intestinal tissues, such as red blood cell infiltration and goblet cell count, lacks quantitative analysis. Can more objective data be provided through image analysis software? |
Response 8: We thank reviewer#1 for his/her insightful comment. We agree that the current description of inflammatory changes, such as red blood cell infiltration and goblet cell count, relies primarily on qualitative histopathological observations and lacks quantitative data. The main outcomes we used to evaluate the purgative effect of C. tiglium were the number of fecal pellets and the water content in the feces. We did not intend to analyze the goblet cell count in this experiment, which is consistent with other studies that also did not perform quantitative analysis, as shown in the references below. Hu et al. Comparative study of the laxative effects of konjac oligosaccharides and konjac glucomannan on loperamide-induced constipation in rats. Food Funct. 2021. DOI: 10.1039/d1fo01237a. Kim et al. Aqueous extracts of Liriope platyphylla induced significant laxative effects on loperamide-induced constipation of SD rats. BMC Complementary and Alternative Medicine 2013, 13:333. http://www.biomedcentral.com/1472-6882/13/333 |
Comments 9: The paper mentions that PMA may lose its activity through gastric acid hydrolysis, but does not cite relevant literature to support this hypothesis. Can the author provide additional experimental or literature evidence to simulate the gastric acid environment in vitro? |
Response 9: We initially assumed that phorbol esters would behave same way of drugs in ester form, which can be hydrolyzed by gastric acid (HCl) or digestive enzymes such as gastric esterases, for example, Aspirin, Enalapril, and Chloramphenicol palmitate. However, after a more in-depth investigation, we have found that phorbol esters are relatively stable in the acidic environment of the stomach but are likely to undergo enzymatic hydrolysis in the intestine or liver. We have revised the sentence from: “It is probable that PMA (and other phorbol esters) can be hydrolyzed in by enzymes the liver or intestine, thus its tumor promotor activity could be lost in the systemic administration of C. tiglium seeds.” (Page 12, lines 353-355). The literature evidence is shown in the references below. Berry D. L., W. M. Bracken, S. M. Fischer, A. Viaje, and T. J. Slaga. 1978. Metabolic conversion of 12-O-tetradecanoylphorbol-13-acetate in adult and newborn mouse skin and mouse liver microsomes. Cancer Res. 38:2301–2306. Mentlein, R. 1986. The tumor promoter 12-O-tetradecanoyl phorbol 13-acetate and regulatory diacylglycerols are substrates for the same carboxylesterase. J. Biol. Chem. 261:7816–7818. Shoyab, M., T. C.Warren, and G. L. Todaro. 1981. Isolation and characterization of an ester hydrolase active on phorbol diesters from murine liver. J. Biol.Chem. 256:12529–12534. |

Reviewer 2 Report (Previous Reviewer 3)
Comments and Suggestions for Authors
1. In response to the reviewer's comment 1, the Authors wrote "Thank you very much for going through this point. Upon verification, we have found that it aligns with your comments. We have calculated the amounts of crotonic acid and PMA and compared them with those in mg/g of CB powder and CA powder, not in mg/g of extract."
However, they forgot to make ALL necessary corrections in the revised manuscript:
Page 8, section 2.5. Determination of the Amounts of Crotonic Acid and PMA: "The amount of PMA in the CB powder was 1.59 ± 0.01 mg/g whereas the amount of PMA in the CA powder was 1.22 ± 0.00 mg/g OF THE EXTRACT."
2. Information about the deuterated solvent used to dissolved the sample should be added also to the section "4.9. Structure Elucidation of DEHP". Currently, this detail is present only in the Supplementary file.
Author Response
Comments 1: In response to the reviewer's comment 1, the Authors wrote "Thank you very much for going through this point. Upon verification, we have found that it aligns with your comments. We have calculated the amounts of crotonic acid and PMA and compared them with those in mg/g of CB powder and CA powder, not in mg/g of extract." However, they forgot to make ALL necessary corrections in the revised manuscript: Page 8, section 2.5. Determination of the Amounts of Crotonic Acid and PMA: "The amount of PMA in the CB powder was 1.59 ± 0.01 mg/g whereas the amount of PMA in the CA powder was 1.22 ± 0.00 mg/g OF THE EXTRACT." |
Response 1: We thank reviewer #2 for raising this issue. We have now revised the sentence to “The amount of PMA in the CB powder was 1.59 ± 0.01 mg/g, whereas the amount of PMA in the CA powder was 1.22 ± 0.00 mg/g.” (page 9, section 2.5) |
Comments 2: Information about the deuterated solvent used to dissolved the sample should be added also to the section "4.9. Structure Elucidation of DEHP". Currently, this detail is present only in the Supplementary file. |
Response 2: We have revised section 4.9. Structure Elucidation of DEHP to include the deuterated solvent used. The revised sentence now is “Pure DEHP was dissolved in CDCl3 for NMR analysis.” (page 18, line 62) |

Round 2
Reviewer 1 Report (Previous Reviewer 2)
Comments and Suggestions for Authors
The author has made the necessary modifications and explanations as per my request, therefore I agree to accept it in its current form.
This manuscript is a resubmission of an earlier submission. The following is a list of the peer review reports and author responses from that submission.
Round 1
Reviewer 1 Report
Comments and Suggestions for Authors
This manuscript titled "Purgative Effect, Acute Toxicity, and Quantification of Phorbol-12-Myristate-13-Acetate and Crotonic Acid in Croton tiglium L. Seed before and after Treatment by Thai Traditional Detoxification Process". In my opinion, it is not suitable for publication in International Journal of Molecular Sciences. The comments for this manuscript are as follows:
1. Many toxic components have been discovered in Croton tiglium L. seed, and this article selects only two components, which are not representative.
2. The dosage selection in the article was not explained in detail. What is the basis for dose selection?
3. Figure 3, three key images were missing.
4. Figure 4, how can we determine the retention time of crotonic acid and PMA without standard reference spectra?
5. Figure 4, the authors did not conduct sufficient analysis on the changes in chemical composition before and after processing.
Author Response
Comments 1: Many toxic components have been discovered in Croton tiglium L. seed, and this article selects only two components, which are not representative. |
Response 1: Thank you for pointing this out. The main objective of this study was not to identify all the compounds discovered in Croton tiglium seeds, some of which were allegedly claimed to be toxic. However, most of them were tested for their toxicity in different methods and systems. We chose to study only PMA and crotonic acid because these compounds, which were found to be tumor promotor, were the cause of the prohibition of the use of C. tiglium seeds in Thai Traditional Medicine. Therefore, we wish to give a scientific-based evidence to prove if the traditional knowledge used to detoxify C. tiglium seeds has any validity. |
Comments 2: The dosage selection in the article was not explained in detail. What is the basis for dose selection? |
Response 2: We calculated the dose of C. tiglium seeds based on the average amount found in Thai traditional medicine recipes from Thai traditional medicine textbooks, where the typical range is 10–60 mg/kg BW. We also included a higher dose level of 100 mg/kg BW. We have added this information to the manuscript. (Page 17, lines 629-632). |
Comments 3: Figure 3, three key images were missing. |
Response 3: Thank you for your observation. This error may have resulted from deformatting due to differences in Microsoft Office on each computer and the conversion from a Word file to a PDF file. We have corrected it. |
Comments 4: Figure 4, how can we determine the retention time of crotonic acid and PMA without standard reference spectra? |
Response 4: The retention times of crotonic acid and PMA were determined and described in sub-section “4.7.1. Determination of the Amounts of Crotonic Acid and PMA” in the paragraph “The HPLC chromatograms of the reference standards, crotonic acid and PMA, showed the retention times of crotonic acid and PMA at 9.45 and 57.55 min, respectively. The peak positions of crotonic acid and PMA in the HPLC chromatograms of the CB and CA extracts were identified by comparing their retention times with those of the reference standards, as well as by spiking crotonic acid and PMA into the extracts (Page 17, lines 654-657). |
Comments 5: Figure 4, the authors did not conduct sufficient analysis on the changes in chemical composition before and after processing. |
Response 5: This study is the first preliminary investigation of the effects of the Thai Traditional Detoxification Process (TDP) on the changes in chemical composition, pharmacological activities, and toxicity of C. tiglium seeds before and after TDP. The objective of this study is not to make a chemical analysis of all compounds before and after the TDP, but to focus on the alteration of the amounts of the compounds (PMA and crotonic acid) that are used as a basis for the prohibition to use in Thai traditional medicine by the health authority of Thailand. As you can see from the HPLC chromatogram of CA and CB, there are hundreds of compounds in the extracts, but only a few of them disappeared or appeared after the TDP. Therefore, our future study will concentrate on the isolation and structure elucidation of these compounds. |
Reviewer 2 Report
Comments and Suggestions for Authors
I reviewed the manuscript entitled Purgative Effect, Acute Toxicity, and Quantification of Phorbol-12-Myristate-13-Acetate and Crotonic Acid in Croton tiglium L. Seed before and after Treatment by Thai Traditional Detoxification Process.
I agree to accept this manuscript after major revision.
1) Abstract: This TDP has never been scientifically proved for its efficacy. This research aims to investigate the effects of TDP on purgative effect and acute toxicity as well as the chemical constituents in C. tiglium seeds before (CB) and after (CA) treatment. C. Tiglium should use its full name, and abbreviations are only necessary if they appear three or more times, as too many abbreviations can confuse readers. Please review and revise the entire text according to this principle.
2) C. tiglium ripe seeds contain several anticancer compounds, including alkaloids, flavonoids [11-13] and diterpenes such as 12-O-tiglylphorbol-13-(2-methyl)butyrate, 12-O-acetylphorbol-13-isobutyrate, 12-O-benzoylphorbol-13-(2-methyl)butyrate, 12-O-tiglyl-7-oxo-5-ene-phorbol-13-(2-methylbutyrate) and 13-O-(2-methyl) butyryl-4-deoxy-4α-phorbol [8]. Diterpenes should change to diterpenoids. Diterpenes: Refers to a basic hydrocarbon skeleton composed of four isoprene units (C20), without other functional groups or oxidative modifications. For example, the parent nucleus structure of phytol or abietic acid in plants. Diterpenoids: are oxidized derivatives of diterpenes, typically containing functional groups such as ester, hydroxyl, and ketone groups. Phorbol esters (such as the compounds mentioned in the article) belong to the typical class of diterpenes, as their structure contains a multi-oxidized ring system and esterification side chains (such as 12-O-acylation, 13-O-acylation, etc.). The compounds listed in the article, such as 12-O-tigylphorbol-13- (2-methyl) butyrate, are all derivatives of phorbol esters with highly oxidized skeletons and ester bonds, which meet the definition of diterpenoids.
3) which received 2,000 mg/kg BW of CB and CA powder, BW cannot use abbreviations directly. The full names should be listed, and abbreviations should be used in parentheses. When using them again, abbreviations should be used directly. That is, body weight (BW).
4) Table 1. Body weight, Food consumption and Water intake, Why are there no SEM for the results of these indicators? The experimental results should be expressed as mean ±SEM, and the differences in the results should be marked with letters (a, b).
5) Table 2. Week0 should change to Week 0.Week1 and Week2 also needs to be modified. The same issue also exists in other tables, please check and modify.
6) Table 4. WBC (103/ul) ul should change to mL. HGB (g/dl) dl should change to dL. Table 4 and Table 5, abbreviations in the table should be listed with their full names in the footnotes.
7) and water content in feces after 8 hours of the experiment were not significantly different among most groups, 8 hours should change to 8 h. day should change to d. Use international units instead of words. Check and revise the entire text.
8) The decrease in PMA could be due to the hydrolysis of PMA to phorbol. This method not only reduced the amount of crotonic acid and PMA but also produced another compound that appeared in the chromatogram of CA extract, with a retention time of 28 min. However, this compound has not yet been identified. Given the condition of the detoxification process, this compound may well be a product of hydrolysis of PMA. Therefore, further study is needed to elucidate the structure of this compound. UHPLC-MS is an ideal technique for determining whether PMA is hydrolyzed into phorbol. Can the author explain why UHPLC-MS is not used? Can you supplement this experiment? If not, it should also be stated in the conclusion section that this is one of the limitations of this study.
9) C. tiglium seeds (Figure 5.(D)) were collected from Chumphon Province, Thailand. The voucher specimen (voucher no MSU.PH-EUP-C1) was deposited at the herbarium of the Faculty of Pharmacy, Mahasarakham University, Maha Sarakham, Thailand. There is a lack of information about the identifier, it is best to add coordinate information such as latitude and longitude, as well as altitude.
10) Figure 5. Photograph of C. tiglium L., leaves (A) flower (B) fresh fruit. (C) dry fruit and seed (D) The singular and plural numbers should be consistent. They should be changed to flowers (B) fresh fruits. (C) dry fruits and seeds (D)
11) 4.3. Treatment of C. tiglium seeds by Thai Traditional Detoxification Process (TDP) The first letter of the actual word in the secondary title needs to be capitalized. Therefore, seeds should change to Seeds. Check and revise the entire text.
12) The study only used female Wistar rats for acute toxicity testing. Has the impact of gender differences on the results been considered? Is there a plan to include male rats in future studies?
13) Is there a more detailed basis to support the selection of doses of 300 mg/kg and 2000 mg/kg in acute toxicity testing? For example, did you refer to previous research or pre experimental data?
14) Is the boiling time and temperature standardized in the specific steps of Thailand's traditional detoxification process (TDP)? Is there any data to support the impact of these parameters on detoxification effectiveness?
15) The study mentioned that TDP completely eliminated crotonic acid, but PMA only reduced by about 23%. Is this partial reduction sufficient to support the effectiveness of TDP? Do we need a more sensitive detection method?
16) The research conclusion suggests a reassessment of the toxicity classification of croton seeds, but does not specify their safe dosage range in traditional medicine. Do you have any relevant suggestions?
17) The actual use effect of TDP treated seeds in traditional formulations was not mentioned in the study. Is there clinical or traditional usage data to support it?
18) I have read all the references and found some issues. Refs 1 and 6, the first letter of euphorbiaceae and croton need to be capitalized. Refs 12 and 13, the first letter of each actual word in the journal name also needs to be capitalized. Refs 13 and 15, missing information on the volume. Please review and revise all literature according to the requirements of the journal.
19) The biggest problem with this study is many details need to be modified and improved. Too many issues can make people feel that the author's attitude is not rigorous.The author must take them seriously and make necessary revisions.
Author Response
Comments 1: Abstract: This TDP has never been scientifically proved for its efficacy. This research aims to investigate the effects of TDP on purgative effect and acute toxicity as well as the chemical constituents in C. tiglium seeds before (CB) and after (CA) treatment. C. Tiglium should use its full name, and abbreviations are only necessary if they appear three or more times, as too many abbreviations can confuse readers. Please review and revise the entire text according to this principle. |
Response 1: Thank you for pointing this out. We follow the standard norm of using the names of biological materials in publications. In this case, the full name of the plant, Croton tiglium, is used when it is first mentioned. After that, we use the abbreviation of the genus (C. for Croton) |
Comments 2: C. tiglium ripe seeds contain several anticancer compounds, including alkaloids, flavonoids [11-13] and diterpenes such as 12-O-tiglylphorbol-13-(2-methyl)butyrate, 12-O-acetylphorbol-13-isobutyrate, 12-O-benzoylphorbol-13-(2-methyl)butyrate, 12-O-tiglyl-7-oxo-5-ene-phorbol-13-(2-methylbutyrate) and 13-O-(2-methyl) butyryl-4-deoxy-4α-phorbol [8]. Diterpenes should change to diterpenoids. Diterpenes: Refers to a basic hydrocarbon skeleton composed of four isoprene units (C20), without other functional groups or oxidative modifications. For example, the parent nucleus structure of phytol or abietic acid in plants. Diterpenoids: are oxidized derivatives of diterpenes, typically containing functional groups such as ester, hydroxyl, and ketone groups. Phorbol esters (such as the compounds mentioned in the article) belong to the typical class of diterpenes, as their structure contains a multi-oxidized ring system and esterification side chains (such as 12-O-acylation, 13-O-acylation, etc.). The compounds listed in the article, such as 12-O-tigylphorbol-13- (2-methyl) butyrate, are all derivatives of phorbol esters with highly oxidized skeletons and ester bonds, which meet the definition of diterpenoids. |
Response 2: Thank you for a valuable comment. We will revise the term in the body text of the manuscript accordingly. |
Comments 3: which received 2,000 mg/kg BW of CB and CA powder, BW cannot use abbreviations directly. The full names should be listed, and abbreviations should be used in parentheses. When using them again, abbreviations should be used directly. That is, body weight (BW). |
Response 3: Thank you for your suggestion. We have given the full names “body weight” and abbreviations when we first mentioned them in the Abstract. |
Comments 4: Table 1. Body weight, Food consumption and Water intake, Why are there no SEM for the results of these indicators? The experimental results should be expressed as mean ±SEM, and the differences in the results should be marked with letters (a, b). |
Response 4: Table 1 presents the results of a sighting study, an initial test conducted to determine the initial dose of the substance to be used in the main study for toxicity testing. In the sighting study, rats were divided into five groups, with only one rat per group. That is the reason why there is no SEM. |
Comments 5: Table 2. Week0 should change to Week 0. Week1 and Week2 also needs to be modified. The same issue also exists in other tables, please check and modify. |
Response 5: Thank you for pointing this out. We have thoroughly examined the entire manuscript. |
Comments 6: Table 4. WBC (103/ul) ul should change to mL. HGB (g/dl) dl should change to dL. Table 4 and Table 5, abbreviations in the table should be listed with their full names in the footnotes. |
Response 6: Agree. We have made the corrections according to your suggestion. |
Comments 7: and water content in feces after 8 hours of the experiment were not significantly different among most groups, 8 hours should change to 8 h. day should change to d. Use international units instead of words. Check and revise the entire text. |
Response 7: Agree. We have made the corrections according to your suggestion. |
Comments 8: The decrease in PMA could be due to the hydrolysis of PMA to phorbol. This method not only reduced the amount of crotonic acid and PMA but also produced another compound that appeared in the chromatogram of CA extract, with a retention time of 28 min. However, this compound has not yet been identified. Given the condition of the detoxification process, this compound may well be a product of hydrolysis of PMA. Therefore, further study is needed to elucidate the structure of this compound. UHPLC-MS is an ideal technique for determining whether PMA is hydrolyzed into phorbol. Can the author explain why UHPLC-MS is not used? Can you supplement this experiment? If not, it should also be stated in the conclusion section that this is one o wasf the limitations of this study. |
Response 8: This issue has raised our curiosity as well. Therefore, we have isolated and purified this compound in sufficient quantity for NMR analysis. Fortunately, we have succeeded elucidating its structure as di-(2-ethylhexyl) phthalate, using 1D and 2D NMR and HRMS just after we had submitted the manuscript. Therefore, we have already included this information in the discussion (sections “2.6 Identification of di-(2-ethylhexyl)phthalate (DEHP)) and experimental (section 4.8 Isolation of di-(2-ethylhexyl)phthalate (DEHP) 4.9. Structure Elucidation of DEHP). The structure of di-(2-ethylhexyl)phthalate (Figure S1), its 1H and 13CNMR data (Table S1), 1D and 2D NMR spectra (Figures S2-7) and HRMS (Figure S8) are in “Supplementary Materials”. |
Comments 9: C. tiglium seeds (Figure 5.(D)) were collected from Chumphon Province, Thailand. The voucher specimen (voucher no MSU.PH-EUP-C1) was deposited at the herbarium of the Faculty of Pharmacy, Mahasarakham University, Maha Sarakham, Thailand. There is a lack of information about the identifier, it is best to add coordinate information such as latitude and longitude, as well as altitude. |
Response 9: We have added the identifier of authentic plant and the coordinations ( latitudes and longitudes) of the location in Chumphon Province in the manuscript. |
Comments 10: Figure 5. Photograph of C. tiglium L., leaves (A) flower (B) fresh fruit. (C) dry fruit and seed (D) The singular and plural numbers should be consistent. They should be changed to flowers (B) fresh fruits. (C) dry fruits and seeds (D) |
Response 10: Thank you for your valuable suggestion. We have revised the manuscript as advised. |
Comments 11: 4.3. Treatment of C. tiglium seeds by Thai Traditional Detoxification Process (TDP) The first letter of the actual word in the secondary title needs to be capitalized. Therefore, seeds should change to Seeds. Check and revise the entire text. |
Response 11: Thank you for your valuable suggestion. We have revised the manuscript as advised. |
Comments 12: The study only used female Wistar rats for acute toxicity testing. Has the impact of gender differences on the results been considered? Is there a plan to include male rats in future studies? |
Response 12: The OECD (Organisation for Economic Co-operation and Development) Guidelines for the Testing of Chemicals, particularly for acute oral toxicity (e.g., Test Guidelines 420, 423, and 425), generally recommend the use of female rats for acute toxicity testing. There are several key reasons for this preference: 1. Increased Sensitivity: Literature surveys and historical data from conventional LD50 tests (the older method for acute toxicity) have often shown that female rats are generally slightly more sensitive to the toxic effects of various chemicals compared to males. This means that if a substance is toxic, it's more likely to induce signs of toxicity or mortality in females at lower doses. By using the more sensitive sex, the test provides a more conservative and protective assessment of the chemical's hazard, ensuring that potential risks are not underestimated. 2. Animal Welfare Considerations (Reduction): The OECD guidelines are strongly committed to the "3Rs" principle: Replace, Reduce, and Refine animal use in testing. Reduction: By typically using only one sex (females) in the main study, the number of animals required for the test is significantly reduced compared to testing both sexes. If there is strong evidence or prior knowledge indicating that males might be more sensitive for a specific chemical, then males can be used instead, but this requires justification. The aim is to obtain sufficient information for classification and labeling with the fewest animals possible. 3. Efficiency and Simplicity: Using a single sex simplifies the experimental design and reduces the variability that might arise from sex-dependent differences in toxicokinetics (how the body processes the chemical) or toxicodynamics (how the chemical affects the body). This makes the results more straightforward to interpret for regulatory purposes. 4. Globally Harmonised System (GHS) Classification: The purpose of acute toxicity tests under OECD guidelines is to classify chemicals according to the Globally Harmonised System of Classification and Labelling of Chemicals (GHS). The GHS is designed to provide clear and consistent information on hazardous chemicals. By using the generally more sensitive sex, the classification arrived at is likely to be adequately protective for both sexes and for humans. |
Comments 13: Is there a more detailed basis to support the selection of doses of 300 mg/kg and 2000 mg/kg in acute toxicity testing? For example, did you refer to previous research or pre experimental data? |
Response 13: We calculated the dose of C. tiglium seeds based on the average amount found in Thai traditional medicine recipes from Thai traditional medicine textbooks, where the typical range is 10–60 mg/kg BW (Page 17, lines 629-632). These doses are relatively safe; therefore, we are interested in investigating doses higher than the normal level, ranging from 5 times the typical dose up to the maximum dose that can be administered to rats. |
Comments 14: Is the boiling time and temperature standardized in the specific steps of Thailand's traditional detoxification process (TDP)? Is there any data to support the impact of these parameters on detoxification effectiveness? |
Response 14: This is the first scientific study to evaluate the effectiveness of the Thai Traditional Detoxification Process (TDP), and no prior data are available. Therefore, we strictly followed the procedures as described in traditional Thai medical texts. We used the same conditions (boiling time and temperature, etc.) as it is traditionally used for the detoxification process. |
Comments 15: The study mentioned that TDP completely eliminated crotonic acid, but PMA only reduced by about 23%. Is this partial reduction sufficient to support the effectiveness of TDP? Do we need a more sensitive detection method? |
Response 15: As mentioned above, this is the first scientific study to evaluate the effectiveness of the Thai Traditional Detoxification Process (TDP). The results support that TDP can reduce the laxative potency of C. tiglium seed powder and show no acute toxicity, both before and after the process. The findings also reveal that TDP can completely eliminate crotonic acid and partially reduce the amount of PMA, which paves the way for further studies aimed at enhancing the reduction of PMA in the future. Both crotonic acid and PMA are toxic compounds. However, PMA is generally considered more toxic than crotonic acid. PMA is a potent activator of protein kinase C and a tumor promoter, which can lead to inflammation and tissue damage. On the other hand, while crotonic acid is a corrosive chemical, primarily causes irritation and burns. Since crotonic acid exists in Croton tiglium seeds in a minute quantity, its elimination is easier (through decarboxylation) while PMA needs to undergo hydrolysis to lose its toxicity. However, the concentration of PMA is much higher in C. tiglium seed oil than in the seeds themselves and thus the toxicity of the seeds is not as severe as that of the seed oil. This can be supported by the fact that Chinese Traditional Medicine uses C. tiglium seeds without treatment. Therefore, the Thai Traditional Detoxification process must be effective enough to decrease a toxicity of PMA present in C. tiglium seeds used in the traditional Thai recipes. |
Comments 16: The research conclusion suggests a reassessment of the toxicity classification of croton seeds, but does not specify their safe dosage range in traditional medicine. Do you have any relevant suggestions? |
Response 16: We calculated the dose of C. tiglium seeds based on the average amount found in Thai traditional medicine recipes from Thai traditional medicine textbooks, where the typical range is 10–60 mg/kg BW (Page 17, lines 629-632).. These doses are relatively safe when compared to this and previous studies. The toxicity evaluation was intended to prove that the use of C. tiglium seeds after being treated with TDP is safe and should not prohibit their use in Thai Traditional medicine recipes/formulations. |
Comments 17: The actual use effect of TDP treated seeds in traditional formulations was not mentioned in the study. Is there clinical or traditional usage data to support it? |
Response 17: As mentioned in the introduction. Although C. tiglium seeds had been used in Thai traditional medicine (TTM) for a long time, they have been banned to use as an ingredient in TTM in Thailand since 1978. There are numerous Thai traditional recipes documented in official Thai medical texts that include C. tiglium seeds as a component, and these formulations have been used in actual practice before 1978. |
Comments 18: I have read all the references and found some issues. Refs 1 and 6, the first letter of euphorbiaceae and croton need to be capitalized. Refs 12 and 13, the first letter of each actual word in the journal name also needs to be capitalized. Refs 13 and 15, missing information on the volume. Please review and revise all literature according to the requirements of the journal. |
Response 18: Thank you for your suggestion. We have thoroughly revised the manuscript to to follow the journal’s format. |
Comments 19: The biggest problem with this study is many details need to be modified and improved. Too many issues can make people feel that the author's attitude is not rigorous. The author must take them seriously and make necessary revisions. |
Response 19: We sincerely appreciate your thoughtful and constructive feedback. We have revised the manuscript in accordance with your suggestions. |
Reviewer 3 Report
Comments and Suggestions for Authors
The aim of the reviewed study was to evaluate the effect of a traditional Thai detoxification process on the laxative activity and acute toxicity of Croton tiglium seeds, and on the levels of their selected components. It seems the influence of this detoxication method on the pharmacological properties and composition of croton seeds have not been investigated before. The work is quite interesting, but it still has a number of shortcomings.
- Page 7, lines 189-191 “The amount of crotonic acid in the CB extract was 0.001 mg/g whereas no crotonic acid was detected in the CA extract. The amount of PMA in the CB extract was 1.59 mg/g of extract whereas the amount of PMA in the CA extract was 1.22 mg/g of extract.”
The CB extract could not contain 0.001 mg of crotonic acid per gram. According to the manuscript, the Authors obtained merely 50 mg of the CB extract (page 13, line 375). If so, 50 mg of the CB extract contained 0.00005 mg of crotonic acid, while the LOQ for crotonic acid was 3.27 µg /mL = 0.00327 mg/mL, and the LOD = 0.98 µg /mL (0.00098 mg/mL), as shown in Table 7…. I suppose that the Authors intended to express the content of crotonic acid (and possibly also of PMA) as mg/gram of seeds, or some other mistake occurred.
- No SD value was shown for crotonic acid and PMA contents.
- It would be better if the Authors had tried to develop a more effective method of extraction. The applied one step extraction could not allow for a very precise determination of the levels of analytes in the seeds. Nevertheless, it was probably still sufficient to compare the composition of both samples.
- Page 12, the section “4.4. Preparation of Croton Seeds”. There is no information, how the croton seeds were actually powdered. In a mortar, or some kind of mill was applied? If they were ground in a laboratory mill, its manufacturer and model should be described.
- Page 15, section 4.7.1, lines 452-454 “The amounts of crotonic acid and PMA in CB and CA extracts were quantified by HPLC method. CB and CA were separately dissolved in methanol and subjected to HPLC for qualitative and quantitative analysis.”
I think the volume of methanol should be described.
- Page 15, section 4.7.1. What was the temperature of the chromatographic column?
- Page 12, lines 335-345 “C. tiglium seeds are treated before being used in traditional medicine not just in Thailand. The treatment of C. tiglium seeds is also used in India, however, the method of treatment is completely different. The method used in India consists of wrapping dried C. tiglium seeds in a clean white cloth to form a bolus which was soaked in a cow dung solution in a mud pot and boiled, and then washed with water. Then the seeds were treated with cow’s urine, followed by lemon juice. Subsequently, the seeds were washed with water and removed from the outer skin and cotyledons and finally fried with ghee. The study to determine the amount of phorbol and fatty acids revealed that there was a significant reduction of phorbol from 5.18 to 3.86 %. The amounts of saturated fatty acids like arachidic acid, behenic acid, stearic acid and palmitic acid were also decreased whereas all the unsaturated fatty acids like oleic acid and linoleic acid seemed to be unaltered [29].”
I am afraid this is an oversimplified presentation of the method(s) of the treatment of croton seeds used in India. The Authors described only a (probably) the most “extreme” Indian detoxification method, while other, more “performer-friendly”, methods do exist. They are described, in more or less detail, in the following publications:
Pal, P. K., Nandi, M. K., & Singh, N. K. (2014). Detoxification of Croton tiglium L. seeds by Ayurvedic process of Śodhana. Ancient Science of Life, 33(3), 157-161.
Meena, A. K., Venkataraman, P., Singh, R., Ganji, K., MK, C., & Srikanth, N. (2021). Evaluation of the role of Shodhana (Purification) process in Croton tiglium seeds for reduction of toxic content. International Journal of Ayurvedic Medicine, 12(3), 565-75.
Jamadagni, P., Ranade, A., Bharsakale, S., Chougule, S., Jamadagni, S., Pawar, S., ... & Gurav, A. (2023). Impact of Shodhana an Ayurvedic purification process on cytotoxicity and mutagenicity of Croton tiglium Linn. Journal of Ayurveda and Integrative Medicine, 14(2), 100710.
Comments on the Quality of English LanguageThe article should be reread to detect possible stylistic, spelling or grammatic errors. E.g., the following sentences are not very good in terms of style:
"The concentration ranges of crotonic acid and PMA were prepared and analyzed by
HPLC in the same conditions as described above. The calibration curve between the concentration of PMA and the area under the curve was plotted and calculated for linear regression."
Author Response
Comments 1: Page 7, lines 189-191 “The amount of crotonic acid in the CB extract was 0.001 mg/g whereas no crotonic acid was detected in the CA extract. The amount of PMA in the CB extract was 1.59 mg/g of extract whereas the amount of PMA in the CA extract was 1.22 mg/g of extract.” The CB extract could not contain 0.001 mg of crotonic acid per gram. According to the manuscript, the Authors obtained merely 50 mg of the CB extract (page 13, line 375). If so, 50 mg of the CB extract contained 0.00005 mg of crotonic acid, while the LOQ for crotonic acid was 3.27 µg /mL = 0.00327 mg/mL, and the LOD = 0.98 µg /mL (0.00098 mg/mL), as shown in Table 7…. I suppose that the Authors intended to express the content of crotonic acid (and possibly also of PMA) as mg/gram of seeds, or some other mistake occurred. |
Response 1: Thank you very much for going through this point. Upon verification, we have found that it aligns with your comments. We have calculated the amounts of crotonic acid and PMA and compared them with those in mg/g of CB powder and CA powder, not in mg/g of extract. |
Comments 2: No SD value was shown for crotonic acid and PMA contents. |
Response 2: We have made the corrections according to your suggestion by giving the SD values for crotonic acid and PMA contents in 2.5 Determination of the Amounts of Crotonic Acid and PMA. |
Comments 3: It would be better if the Authors had tried to develop a more effective method of extraction. The applied one step extraction could not allow for a very precise determination of the levels of analytes in the seeds. Nevertheless, it was probably still sufficient to compare the composition of both samples. |
Response 3: Thank you for pointing this out. The development of methods of extraction is not the objective of the present work. However, we consider to study the different methods in our future work. |
Comments 4: Page 12, the section “4.4. Preparation of Croton Seeds”. There is no information, how the croton seeds were actually powdered. In a mortar, or some kind of mill was applied? If they were ground in a laboratory mill, its manufacturer and model should be described. |
Response 4: C. tiglium seed powders were ground using the mill. We have added this information to the method. “C. tiglium seeds before treatment (CB) and after treatment (CA) by TDP were ground, using a laboratory mill (Spring Green Evolution, Bangkok, Thailand) to a fine powder and kept in a well-closed container at -20 oC until use.” |
Comments 5: Page 15, section 4.7.1, lines 452-454 “The amounts of crotonic acid and PMA in CB and CA extracts were quantified by HPLC method. CB and CA were separately dissolved in methanol and subjected to HPLC for qualitative and quantitative analysis.” I think the volume of methanol should be described. |
Response 5: The volume of methanol has been added to the extraction method. |
Comments 6: Page 15, section 4.7.1. What was the temperature of the chromatographic column? |
Response 6: The chromatographic column was operated at room temperature. (Page 17, line 656) |
Comments 7: Page 12, lines 335-345 “C. tiglium seeds are treated before being used in traditional medicine not just in Thailand. The treatment of C. tiglium seeds is also used in India, however, the method of treatment is completely different. The method used in India consists of wrapping dried C. tiglium seeds in a clean white cloth to form a bolus which was soaked in a cow dung solution in a mud pot and boiled, and then washed with water. Then the seeds were treated with cow’s urine, followed by lemon juice. Subsequently, the seeds were washed with water and removed from the outer skin and cotyledons and finally fried with ghee. The study to determine the amount of phorbol and fatty acids revealed that there was a significant reduction of phorbol from 5.18 to 3.86 %. The amounts of saturated fatty acids like arachidic acid, behenic acid, stearic acid and palmitic acid were also decreased whereas all the unsaturated fatty acids like oleic acid and linoleic acid seemed to be unaltered [29].” I am afraid this is an oversimplified presentation of the method(s) of the treatment of croton seeds used in India. The Authors described only a (probably) the most “extreme” Indian detoxification method, while other, more “performer-friendly”, methods do exist. They are described, in more or less detail, in the following publications: Pal, P. K., Nandi, M. K., & Singh, N. K. (2014). Detoxification of Croton tiglium L. seeds by Ayurvedic process of Śodhana. Ancient Science of Life, 33(3), 157-161. Meena, A. K., Venkataraman, P., Singh, R., Ganji, K., MK, C., & Srikanth, N. (2021). Evaluation of the role of Shodhana (Purification) process in Croton tiglium seeds for reduction of toxic content. International Journal of Ayurvedic Medicine, 12(3), 565-75. Jamadagni, P., Ranade, A., Bharsakale, S., Chougule, S., Jamadagni, S., Pawar, S., ... & Gurav, A. (2023). Impact of Shodhana an Ayurvedic purification process on cytotoxicity and mutagenicity of Croton tiglium Linn. Journal of Ayurveda and Integrative Medicine, 14(2), 100710. |
Response 7: We have incorporated the additional methods of croton seed treatment into the manuscript, as per your suggestion. |
4. Response to Comments on the Quality of English Language |
Point 1: The article should be reread to detect possible stylistic, spelling or grammatic errors. E.g., the following sentences are not very good in terms of style: "The concentration ranges of crotonic acid and PMA were prepared and analyzed by |
Response 1: Thank you for your valuable and sincere suggestions. We have gone through the whole text to rephrase and revise the English language to raise the standard of the manuscript. |
Round 2
Reviewer 2 Report
Comments and Suggestions for Authors
The author has made revisions according to my suggestions and answered my questions. Therefore, I agree to accept it in its current form.